# EXTREME VALUE $k$-MEANS CLUSTERING

## ABSTRACT

Clustering is the central task in unsupervised learning and data mining. $k$-means is one of the most widely used clustering algorithms. Unfortunately, it is generally non-trivial to extend $k$-means to cluster data samples beyond Gaussian distribution, particularly, the clusters with non-convex shape (Beliakov & King, 2006). To this end, we introduce Extreme Value Theory (EVT) to improve the clustering ability of $k$-means. Particularly, the Euclidean space was transformed into a unique probability space denoted as *extreme value space* by EVT. We thus propose a novel algorithm called Extreme Value $k$-means (EV $k$-means), including GEV $k$-means and GPD $k$-means. Besides, we also introduce the tricks to accelerate Euclidean distance computation in improving the computational efficiency of classical $k$-means. Furthermore, our EV $k$-means is extended to an online version, i.e., online Extreme Value $k$-means, in utilizing the Mini Batch $k$-means to cluster streaming data. Extensive experiments are conducted to validate our EV $k$-means and online EV $k$-means on synthetic datasets and real datasets. Experimental results show that our algorithms outperform competitors in most cases.

## 1 INTRODUCTION

Clustering is a fundamental and important task in the unsupervised learning (Jain, 2010; Rui Xu & Wunsch, 2005). It aims at clustering data samples of high similarity into the same cluster. The most well-known clustering algorithm is the $k$-means, whose objective is to minimize the sum of squared distances to their closest centroids. $k$-means has been extensively studied in the literature, and some heuristics have been proposed to approximate it (Jain, 2010; Dubes & Jain, 1988). The most famous one is Lloyd's algorithm (Lloyd, 1982).

The $k$-means algorithm is widely used due to its simplicity, ease of use, geometric intuition (Bottesch et al., 2016). Unfortunately, its bottleneck is that computational complexity reaches $O(nkd)$ (Rui Xu & Wunsch, 2005), since it requires computing the Euclidean distances between all samples and all centroids. The data is embedded in the Euclidean space (Stemmer & Kaplan, 2018), which causes the failure on clustering non-convex clusters (Beliakov & King, 2006). Even worse, $k$-means is highly sensitive to the initial centroids, which usually are randomly initialized. Thus, it is quite possible that the objective of $k$-means converges to a local minimum, which causes the instability of $k$-means, and is less desirable in practice. Despite a stable version – $k$-means++ (Arthur & Vassilvitskii, 2007) gives a more stable initialization, fundamentally it is still non-trivial to extend $k$-means in clustering data samples of non-convex shape.

To solve these problems, this paper improves the clustering ability of $k$-means by measuring the similarity between samples and centroids by EVT (Coles et al., 2001). In particular, we consider the generalized extreme value (GEV) (Jenkinson, 1955) distribution or generalized Pareto distribution (GPD) (Pickands III et al., 1975; DuMouchel, 1975) to transform the Euclidean space into a probability space defined as, *extreme value space*. GEV and GPD are employed to model the maximum distance and output the probability that a distance is an extreme value, which indicates the similarity of a sample to a centroid. Further, we adopt the Block Maxima Method (BMM) (Gumbel, 2012) to choose the maximal distance for helping GEV fit the data. The Peaks-Over-Thresh (POT) method (Leadbetter, 1991) is utilized to model the excess of distance exceeding the threshold, and thus very useful in fitting the data for GPD.

Formally, since both GEV and GPD can measure the similarity of samples and centroids, they can be directly utilized in $k$-means, i.e., GEV $k$-means and GPD $k$-means, which are uniformly

called Extreme Value $k$-means (EV $k$-means) algorithm. In contrast to $k$-means, EV $k$-means is a probability-based clustering algorithm that clusters samples according to the probability output from GEV or GPD. Furthermore, to accelerate the computation of Euclidean distance, We expand the samples and the centroids into two tensors of the same shape, and then accelerate with the high performance parallel computing of GPU.

For clustering steaming data, we propose online Extreme Value $k$-means based on Mini Batch $k$-means (Sculley, 2010). When fit the GEV distribution, we use mini batch data as a block. For the fitting of GPD, we dynamically update the threshold. The parameters of GEV or GPD are learned by stochastic gradient descent (SGD) (LeCun et al., 1998).

The main contributions are described as follows. (1) This paper utilizes EVT to improve $k$-means in addressing the problem of clustering data of non-convex shape. We thus propose the novel Extreme Value $k$-means, including GEV $k$-means and GPD $k$-means. A method for accelerating Euclidean distance computation has also been proposed to solve the bottleneck of $k$-means. (2) Under the strong theoretical support provided by EVT, we use GEV and GPD to transform Euclidean space into extreme value space, and measure the similarity between samples and centroids. (3) Based on Mini Batch $k$-means, We propose online Extreme value $k$-means for clustering streaming data, which can learn the parameters of GEV and GPD online. We corroborate the effectiveness of EV $k$-means and online EV $k$-means by conducting experiments on synthetic datasets and real datasets. Experimental results show that EV $k$-means and online EV $k$-means significantly outperform compared algorithms consistently across all experimented datasets.

## 2    RELATED WORKS

$k$-means and EVT have been extensively studied in the literature in many aspects (Jain, 2010; Rui Xu & Wunsch, 2005; De Haan & Ferreira, 2007). Previous work on $k$-means focused on the following aspects, such as determining the optimal $k$, initializing the centroids, and accelerating $k$-means. Bandyopadhyay & Maulik (2002); Lin et al. (2005); Van der Merwe & Engelbrecht (2003); Omran et al. (2005) propose to select the optimal $k$ value based on the genetic algorithm. Initializing the centroids is a hot issue in $k$-means (Celebi et al., 2013). $k$-means++ (Arthur & Vassilvitskii, 2007) is the most popular initialization scheme. Katsavounidis et al. (1994); Khan & Ahmad (2004); Redmond & Heneghan (2007) proposed density-based initial centroid selection method, that is, selecting the initial cluster center according to the density distribution of the samples. Recently, Bachem et al. (2016) propose using Markov chain Monte Carlo to accelerate $k$-means++ sampling. There is also a lot of work focused on solving the computational complexity of $k$-means. Hamerly (2010) argued that using triangle inequality can accelerate $k$-means. Sinha (2018) showed that randomly sparse the original data matrix can significantly speed up the computation of Euclidean distance. EVT is widely used in many area, such as natural phenomena, finance, and traffic prediction. In recent years, EVT has many applications in the field of machine learning. However, far too little attention has been paid to the combination of $k$-means and EVT. Li et al. (2012) proposes using generalized extreme value distribution for feature learning based on $k$-means. However, our method is significant different from this method. First, they compute the squared distance from a point to the nearest centroid and form a GEV regarding to each point, while we compute the distance from a centroid all data points and then fit the GEV or GPD regarding to each centroid. Second, their algorithm adds the likelihood function as a penalty term into the objective function of $k$-means, but our algorithm changes the objective function by fitting the GEV or GPD for each centroid and assign the data point to the one with the highest probabilities they belong to. Finally, this paper also presents GPD $k$-means and online Extreme Value $k$-means, which is not stated in this paper.

## 3    PRELIMINARIES

### 3.1    $k$-MEANS CLUSTERING ALGORITHM

Denote $\mathcal{X} = \{\boldsymbol{x}_1, \boldsymbol{x}_2, \ldots, \boldsymbol{x}_n\} \subseteq \mathbb{R}^d$ as the dataset and $\mathcal{C} = \{C_1, C_2, \ldots, C_k\}$ as a partition of $\mathcal{X}$ satisfying $C_i \cap C_j = \varnothing, i \neq j$. Let $\Theta = \{\boldsymbol{\mu}_1, \boldsymbol{\mu}_2, \ldots, \boldsymbol{\mu}_k\}$ with $\boldsymbol{\mu}_i \subseteq \mathbb{R}^d$ be the centroid of cluster

$C_i, i = 1, 2, \ldots, k$, that is, $\boldsymbol{\mu}_i = \frac{1}{|C_i|} \sum_{\boldsymbol{x} \in C_i} \boldsymbol{x}$. The sum squared error is defined as

$$J(\mathcal{C}; \Theta) = \sum_{i=1}^{k} \sum_{\boldsymbol{x} \in C_i} \|\boldsymbol{x} - \boldsymbol{\mu}_i\|_2^2 \tag{1}$$

Eq. (1) indicates that the smaller $J$ is, the higher degree of closeness between the samples and their centroids in the clusters, so the similarity of the samples in the clusters is higher. To find the global minimum of Eq. 1, we need to compute all possible cluster partitions, so $k$-means is an NP-hard problem (Aloise et al., 2009). Lloyd's algorithm (Lloyd, 1982) uses a greedy strategy to approximate the Eq. (1) by iteratively optimizing between assigning cluster labels and updating centroids. Specifically, in assigning cluster labels, a cluster label is assigned to each sample according to the closest centroid. When the centroids are being updated, each centroid is updated to the mean of all samples in the cluster. These two steps loop iteratively until the centroids no longer change.

## 3.2 Extreme Value Theory

In this subsection, we first introduce the statistical aspects of a sample maximum in Extreme Value Theory (EVT), which is a branch of statistics dealing with the stochastic behavior of extreme events found in the tail of probability distribution. Let $X_1, X_2, \ldots, X_n$ be a sample of independent copy of $X$ with distribution $F$. It is theoretically interesting to consider the asymptotic behavior of sample maximum and upper order statistics. More specifically, denote $M_n = \max_{1 \leq i \leq n} X_i$ as the sample maximum, whose distribution is

$$\Pr(M_n \leqslant x) = F^n(x), \quad x \in \mathbb{R}. \tag{2}$$

On the other hand, the upper order statistics of the sample is related to the survival function over a threshold $u$, which is

$$\Pr(X > u + x | X > u) = \frac{\Pr(X > u + x)}{\Pr(X > u)} = \frac{1 - F(u + x)}{1 - F(u)}, \quad x > 0. \tag{3}$$

EVT considers the non-degenerated limit when $n \to \infty$ in Eq.(2) and $u \uparrow x^*$ in Eq.(3) by re-scaling the objects, which is presented as the conditions of the *maximum domain of attraction* for $F$.

**Theorem 3.1 (Fisher-Tippett Theorem** (Fisher & Tippett, 1928)) *A distribution function $F$ satisfis the condition of a maximum domain of attraction: if there exists a constant $\xi \in \mathbb{R}$ and sequences $a_n > 0, b_n, n \in \mathbb{N}$ such that*

$$\lim_{n \to \infty} F^n(a_n x + b_n) = \exp\left\{-(1 + \xi x)^{-1/\xi}\right\}, \quad 1 + \xi x > 0. \tag{4}$$

*The shape parameter $\xi$ is called the extreme value index.*

Theorem 3.1 motivates the *Block Maxima Method* (BMM) (Gumbel, 2012): for the block size $s \in \{1, 2, \ldots, n\}$, divide the sample into $m = \llcorner n/s \lrcorner$ blocks of length $s$. Since the data is independent, each block maxima has distribution $F^s$ and can be approximated by a three-parametric generalized extreme value distribution (GEV) $G_{GEV}(\cdot; \mu, \sigma, \xi)$ when the block size $s$ is large enough and the number of blocks $m$ is sufficient. The class of GEV distributions is defined as

$$G_{GEV}(x; \mu, \sigma, \xi) = \exp\left\{-\left(1 + \xi \frac{x - \mu}{\sigma}\right)^{-1/\xi}\right\}, \quad 1 + \xi \frac{x - \mu}{\sigma} > 0. \tag{5}$$

We treat the case of $\xi = 0$ as the limit of $\xi \to 0$. An equivalent representation of the maximum domain of attraction condition is as follows:

**Theorem 3.2 (Pickands-Balkema-de Haan Theorem** (Balkema & De Haan, 1974)) *A distribution function $F$ satisfies the condition of maximum domain of attraction: if there exists a constant $\xi \in \mathbb{R}$ and a positive function $\sigma(t)$ such that*

$$\lim_{u \uparrow x^*} \frac{1 - F(u + \sigma(u)x)}{1 - F(u)} = (1 + \xi x)^{-1/\xi}, \quad 1 + \xi x > 0. \tag{6}$$

*where $x^*$ denotes the right end-point of the support of $F$.*

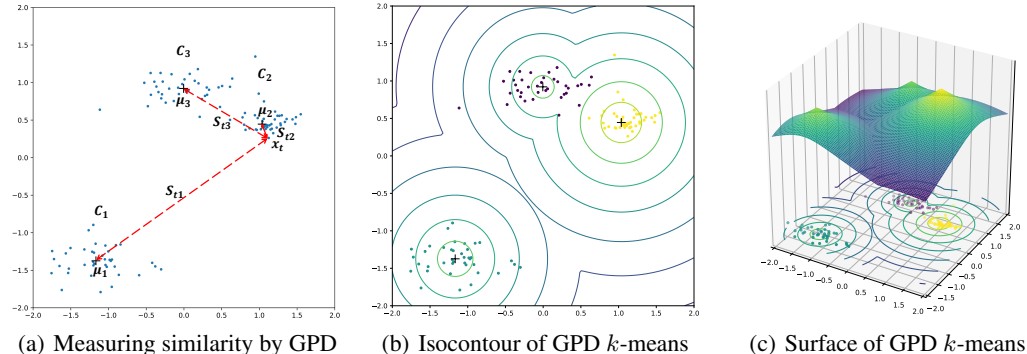

(a) Measuring similarity by GPD     (b) Isocontour of GPD $k$-means     (c) Surface of GPD $k$-means

Figure 1: The clustering results of GPD $k$-means in three isotropic Gaussian blobs. The color of the surface and contour in the figures represent the probability density of GPD. The closer to yellow, the greater the probability density. The closer to blue, the smaller the probability density.

The upper order statistics of a sample usually provides useful information about the tail of the distribution $F$. Then Theorem 3.2 gives rise to an alternative *peak-over-threshold* (POT) approach (Pickands III et al., 1975): given sufficient large threshold $u$ in Eq. (7), we have that, for any $X_i > u$, its conditional distribution can be approximated by a two-parametric generalized Pareto distribution (GPD) $G_{GPD}(\cdot; \sigma, \xi)$, which is defined as

$$G_{GPD}(x; \sigma, \xi) = 1 - \left(1 + \xi \frac{x}{\sigma}\right)^{-1/\xi}, \quad x > 0. \tag{7}$$

Similarly, we treat the case of $\xi = 0$ as the limit of $\xi \to 0$.

The POT approach focuses on the excess over the threshold $u$ to fit the GPD and asymptotically characterize the tail features of the distribution, while the BMM only approximates the GEV distribution when $m$ is large enough. The BMM only uses a very small amount of dataset, and there may be cases where the submaximal value of one block is larger than the maximum value of the other block, which cannot be utilized. In contrast, POT method uses all data beyond the threshold to fit the GPD, making full use of the extreme data. However, there is no winner in theory.

## 4  THE EXTREME VALUE $k$-MEANS ALGORITHM

### 4.1  MEASURING SIMILARITY BY EXTREME VALUE THEORY

Measuring similarity with Euclidean distance is the core step of $k$-means clustering. Similarly, for all clustering algorithms, how to measure the distance (dissimilarity) or similarity between the samples and the centroids is a critical issue (Rui Xu & Wunsch, 2005) as it determines the performance of the algorithm. However, due to the properties of Euclidean distance, $k$-means fails for clustering non-convex clusters. Therefore, this paper proposes to use the EVT to transform the Euclidean space into a probability space called the *extreme value space*. Fig. 1(a) demonstrates measuring similarity by GEV or GPD. The Euclidean distances from $\boldsymbol{\mu}_1$ and $\boldsymbol{\mu}_3$ to $\boldsymbol{x}_t$ is much larger than the Euclidean distances from $\boldsymbol{\mu}_1$ and $\boldsymbol{\mu}_3$ to the most of surrounding samples, i.e. $\|\boldsymbol{x}_t - \boldsymbol{\mu}_1\|_2 \gg \|\boldsymbol{x}_i - \boldsymbol{\mu}_1\|_2, \boldsymbol{x}_i \in C_1$ and $\|\boldsymbol{x}_t - \boldsymbol{\mu}_3\|_2 \gg \|\boldsymbol{x}_i - \boldsymbol{\mu}_3\|_2, \boldsymbol{x}_i \in C_3$. Therefore, $\|\boldsymbol{x}_t - \boldsymbol{\mu}_1\|_2$ and $\|\boldsymbol{x}_t - \boldsymbol{\mu}_3\|_2$ are maximums concerning $\|\boldsymbol{x}_i - \boldsymbol{\mu}_1\|_2, \boldsymbol{x}_i \in C_1$ and $\|\boldsymbol{x}_i - \boldsymbol{\mu}_3\|_2, \boldsymbol{x}_i \in C_3$ with different degree. We want a distribution that can be used to model maximum distance and reflect the probability that a distance is belong to a cluster, which equivalent to the similarity between the sample and the centroid. Obviously, the EVT is a good choice.

As described in Section 3.2, the BMM can be applied to fit the GEV distribution. In order to fit a GEV distribution for each cluster, we first compute the Euclidean distance $d_{ij}$ between $\Theta = \{\boldsymbol{\mu}_1, \boldsymbol{\mu}_2, \ldots, \boldsymbol{\mu}_k\}$ and sample $x_i \in \mathcal{X}$, i.e., $d_{ij} = \|\boldsymbol{x}_i - \boldsymbol{\mu}_j\|_2, i \in \{1, 2, \ldots, n\}, j \in \{1, 2, \ldots, k\}$. For the centroid $\mu_j$, its distances to all samples is denoted by $d_j = \{d_{1j}, d_{2j}, \ldots, d_{nj}\}$. Then we divided them equally into $m$ blocks of size $s = \lfloor \frac{n}{m} \rfloor$ (possibly the last block with no sufficient

observations can be discarded), and then the maximum value of each block is taken to obtain the block maximum sequence $M^j$.

$$M^j = \{M_1, M_2, \ldots, M_m\} \tag{8}$$

We use $M^j$ to estimate the parameters of GEV distributions for cluster $C_j$. We assume the location parameter is zero for the reason that the position of centroids change small in the later stage of clustering. The most commonly used estimating method, maximum likelihood estimation (MLE), is implemented to estimate the two parameters of the GEV. The log likelihood function of GEV is derived from Eq. (5),

$$L_{GEV}(M_j; \sigma_j, \xi_j) = -m \log(\sigma_j) - (1 + \frac{1}{\xi_j}) \sum_{i=1}^{m} \log\left(1 + \xi_j \frac{M_i^j}{\sigma_j}\right) - \sum_{i=1}^{m} \left(1 + \xi_j \frac{M_i^j}{\sigma_j}\right)^{-1/\xi_j}, \ \xi_j \neq 0$$

$$L_{GEV}(M_j; \sigma_j) = -m \log(\sigma_j) - \sum_{i=1}^{m} \frac{M_i^j}{\sigma_j} - \sum_{i=1}^{m} \exp\left(-\frac{M_i^j}{\sigma_j}\right), \ \xi_j = 0$$

$$\tag{9}$$

$1 + \xi_j \frac{M_i^j}{\sigma_j} > 0$ when $\xi_j \neq 0$. We get the estimated value $\hat{\sigma}_j$ and $\hat{\xi}_j$ of $\sigma_j$ and $\xi_j$ by maximizing $L_{GEV}$.

Alternatively, we use the POT method to model the excess of Euclidean distance $d_j$ exceeding threshold $u_j$ for centroid $\boldsymbol{\mu}_j$ and fit the GPD. We first compute the excess that is defined as

$$y^j = d_j - u_j, \ d_j > u_j, \ y^j = \{y_1^j, y_2^j, \ldots, y_{k_j}^j\} \tag{10}$$

where $k_j$ is the total number of observations greater than the threshold $u_j$. Then we also implement MLE to estimate the parameters of the GPD. The log likelihood function of GPD can be derived from Eq. (7),

$$L_{GPD}(y^j; \sigma_j, \xi_j) = k_j \log \xi_j - k_j \log \sigma_j - (1 + \frac{1}{\xi_j}) \sum_{i=1}^{k_j} \log(1 + \frac{\xi_j}{\sigma_j} y_i^j), \quad \xi_j \neq 0$$

$$L_{GPD}(y^j; \sigma_j) = -k_j \log \xi_j - \sigma_j^{-1} \sum_{i=1}^{k_j} y_i^j, \quad \xi_j = 0$$

$$\tag{11}$$

$y_i^j \geqslant 0$ when $\xi_j > 0$ and $0 \leqslant y_i^j \leqslant -\frac{\sigma_j}{\xi_j}$ when $\xi_j < 0$. We get the estimated value $\hat{\sigma}_j$ and $\hat{\xi}_j$ of $\sigma_j$ and $\xi_j$ by maximizing the $L_{GPD}$. Finally, we can obtain the probability that $\boldsymbol{x}_i$ belong to cluster $C_j$ through the GEV and GPD:

$$P_{ij} = 1 - G_{GEV}(d_{ij}; \hat{\sigma}_j, \hat{\xi}_j)$$
$$P_{ij} = 1 - G_{GPD}(d_{ij} - u_j; \hat{\sigma}_j, \hat{\xi}_j)$$

$$\tag{12}$$

### 4.2 Optimization for $k$-means with EVT

The traditional $k$-means clusters samples in view of the closeness to the centroids of clusters. As described in Section 4.1, we can model the distribution classes of GEV and GPD to measure the similarity between the samples and the centroids. Thus we propose GEV $k$-means and GPD $k$-means, which are uniformly called the Extreme Value $k$-means (EV $k$-means) algorithm. In contrast to $k$-means, the proposed EV $k$-means is a probability-based clustering algorithm as it instead clusters samples by the probability output from GEV or GPD. The larger the block size $s$ and the threshold $u$ of BMM and POT, the smaller the deviation of MLE, but the larger the variance of MLE. Conversely, the smaller the block size $s$ and the threshold $u$, the larger the deviation of the MLE, but the smaller the variance of the MLE. How to choose these two hyperparameters has not yet had a standard method, and it is necessary to comprehensively balance the relationship between deviation and variance in practical applications. Therefore, we set the block size by grid search and set threshold adaptively. Specifically, we first set the hyperparameter $\alpha$ to indicate the percentage of excess for all samples. Then we sort $d_j$ from big to small, and the $u$ is set to the $\alpha n$-th of sorted $d_j$.

The algorithm of GEV $k$-means has three steps: Given the dataset $\mathcal{X}$, block size $s$ and $k$ initial centroids (obtained randomly or using $k$-means++ algorithm). During the step of fitting a GEV

distribution, we firstly use BMM to select the maximal sample data $M^j$ for $\boldsymbol{\mu}_j$. Then, we estimate the GEV parameters $\hat{\sigma}_j$ and $\hat{\xi}_j$ by MLE using $M_j$ for $\boldsymbol{\mu}_j$. So each cluster has its own independent GEV distribution. In the assigning labels step, each sample is assigned a cluster label based on the maximum probability, i.e., $\lambda_i = \arg\max_{j \in \{1,2,\ldots,k\}} P_{ij}$. In the updating centroid step, each centroid is updated to the mean of all samples in the cluster, i.e., $\boldsymbol{\mu}_i = \frac{1}{|C_i|} \sum_{\boldsymbol{x} \in C_i} \boldsymbol{x}$. There three steps are iterated until the centroids no longer change.

The algorithm of GPD $k$-means is very similar to GEV $k$-means, except the fitting GPD distribution step. GPD $k$-means use the POT to model the excess $y^j$ of Euclidean distance $d_j$ exceeding threshold $u_j$ and fit the GPD. Fig. 1(b) and Fig. 1(c) show clustering results of GPD $k$-means in three isotropic Gaussian blobs and show that the closer to the centroids, the greater the probability density.

### 4.3 SPEEDING UP

The main bottleneck of the $k$-means is the computation of Euclidean distances for the reason that the Euclidean distances between all samples and all centroids need to be computed. In naïve implementation, double-layer nested for loop is often used to perform operations on the CPU, which is very slow. This paper proposes an accelerated computation method to solve this bottleneck. Firstly, let matrix $\boldsymbol{X} \in \mathbb{R}^{n \times d}$ represents samples consisting of $n$ $d$-dimensional samples, and matrix $\boldsymbol{C} \in \mathbb{R}^{k \times d}$ represents centroids consisting of $k$ $d$-dimensional centroids. Secondly, insert a dimension between the two dimensions of matrix $\boldsymbol{X}$ and copy $\boldsymbol{X}$ along the new dimension to tensor $\mathbf{X}$ with shape of $[n, k, d]$. A similar operation for matrix $\boldsymbol{C}$, adding a new dimension before the first dimension and copy $\boldsymbol{C}$ along the new dimension to tensor $\mathbf{C}$ with shape of $[n, k, d]$. Finally, the Euclidean distances between all samples and all centroids are $\boldsymbol{D}_{i,j} = \|\mathbf{X} - \mathbf{C}\|_2, i \in \{1, 2, \ldots, n\}, j \in \{1, 2, \ldots, k\}$ that can be accelerate with the advantages of GPU parallel computing. The overall Extreme Value $k$-means algorithm is illustrated in Algorithm 1.

### 4.4 ONLINE LEARNING FOR CLUSTERING STREAMING DATA

In the era of Big Data, data is no longer stored in memory, but in the form of streams (Bugdary & Maymon, 2019). Therefore, clustering streaming data is a significant and challenging problem. It is indispensable to design an Extreme Value $k$-menas algorithm that can learn online for clustering streaming data. This paper proposes the online Extreme Value $k$-means for clustering streaming data based on Mini Batch $k$-means (Sculley, 2010). When fit the GEV distribution, we use mini batch data as a block and choose the maximum value for learning the parameters of GEV online. For the fitting of the GPD, the online EV $k$-means can dynamically update the threshold $u$ and learn the parameters of GPD online.

The Online Extreme Value $k$-means algorithm is illustrated in Algorithm 2. The algorithm randomly choose a mini batch contains $b$ samples from the data stream each iteration. On the first iteration, it initializes the parameters of each GEV or GPD, and initializes centroid $\boldsymbol{C}$ on the mini batch.

Then compute the Euclidean distances $\boldsymbol{D}$ using the accelerated computation method we proposed, update $u_j$ to $t\alpha n$-th of sorted $h$, and compute the maximum $M^j$ and excess $y_j$. Because the GEV and GPD parameters have not been updated at the first iteration, so $P_{ij}$ cannot be computed. Therefore, from the second iteration, the algorithm clusters the mini batch samples based on Mini Batch $k$-means. Finally, the negative log-likelihood function of all GEVs or GPDs is summed to obtain $L_s$, and the $L_s$ is minimized by SGD to update the parameters of GEV or GPD, which is equivalent to maximizing $\sum_{j=1}^{k} L_{GEV}(M^j; \sigma_j, \xi_j)$ and $\sum_{j=1}^{k} L_{GPD}(y^j; \sigma_j, \xi_j)$.

## 5 EXPERIMENTS AND RESULTS

### 5.1 EVALUATION METRICS

We evaluate the performance of the clustering algorithm by three widely used metrics, unsupervised clustering accuracy (ACC) (Cai et al., 2010), normalized mutual information (NMI) (Vinh et al., 2010), and adjusted rand index (ARI) (Vinh et al., 2010). Note that the values of ACC and NMI are

---

**Algorithm 1:** Extreme Value $k$-means

---

**Input:** samples $\boldsymbol{X} \in \mathbb{R}^{n \times d}$, number of cluster $k$, block size $s$ for GEV $k$-means, the percentage of excess $\alpha$ for GPD $k$-means

**Output:** clusters $\mathcal{C}$

Initialize centroid $\boldsymbol{C} \in \mathbb{R}^{k \times d}$;

**repeat**

    $C_j = \varnothing,\ 1 \leqslant j \leqslant k$;

    Perform transformation on $\boldsymbol{X}$ and $\boldsymbol{C}$ to obtain $\mathsf{X}$ and $\mathsf{C}$, and then compute $D = \|\mathsf{X} - \mathsf{C}\|_2$;

    **for** $j = 1, 2, \ldots, k$ **do**

        `// GEV` $k$`-means`

        Obtain $M^j$ from $d_{:,j}$ by BMM;

        Estimate the $\hat{\sigma}_j, \hat{\xi}_j$ by MLE on $M^j$;

        `// GPD` $k$`-means`

        Obtain $y^j$ from $D_{:,j}$ by POT;

        Estimate the $\hat{\sigma}_j, \hat{\xi}_j$ by MLE on $y^j$;

    **end**

    **for** $i = 1, 2, \ldots, n$ **do**

        $\lambda_i = \arg \max_{j \in \{1,2,\ldots,k\}} P_{ij}$;

        $C_{\lambda_j} = C_{\lambda_j} \cup x_i$

    **end**

    **for** $j = 1, 2, \ldots, k$ **do**

        $\boldsymbol{\mu}_j = \frac{1}{|C_j|} \sum_{\boldsymbol{x} \in C_j} \boldsymbol{x}$;

    **end**

**until** *centroids no longer change*;

return clusters $\mathcal{C}$;

---

**Algorithm 2:** online Extreme Value $k$-means

---

**Input:** samples $\boldsymbol{X} \in \mathbb{R}^{n \times d}$, number of cluster $k$, mini-batch size $b$, the percentage of excess $\alpha$

**Output:** centroid $\boldsymbol{C}$

Initialize $u_j = 0,\ h = [\,],\ N = 0$ ;

**for** $t = 1, 2, \ldots, n/d$ **do**

    $M \leftarrow$ choose $b$ samples randomly from $\boldsymbol{X}$;

    **if** $t == 1$ **then**

        Initialize centroid $\boldsymbol{C} \in \mathbb{R}^{k \times d}$ and the parameters of GEVs or GPDs;

    **end**

    Perform transformation on $M$ and $\boldsymbol{C}$ to obtain $\mathsf{M}$ and $\mathsf{C}$, and compute the $D = \|\mathsf{M} - \mathsf{C}\|_2$;

    **for** $j=1,2,\ldots,k$ **do**

        `// online GEV` $k$`-means`

        $M^j = \max(\boldsymbol{D}_{:,j})$;

        `// online GPD` $k$`-means`

        $h.append(\boldsymbol{D}_{:,j}[\boldsymbol{D}_{:,j} > u_j])$, Sort $h$ from big to small and set $u_j = h[t\alpha n]$;

        $y^j = \boldsymbol{D}_{:,j}[\boldsymbol{D}_{:,j} > u_j] - u_j$;

    **end**

    **if** $t \geqslant 2$ **then**

        **for** $i = 1, 2, \ldots, b$ **do**

            $\lambda_i = \arg \max_{j \in \{1,2,\ldots,k\}} S_{ij}$;

            $N[\lambda_i] = N[\lambda_i] + 1$;

            $\gamma = \frac{1}{N[\lambda_i]}$;

            $\boldsymbol{C}_{\lambda_i,:} = (1 - \gamma)\boldsymbol{C}_{\lambda_i,:} + \gamma\boldsymbol{M}_{i,:}$;

        **end**

    **end**

    $L_s = -\sum_{j=1}^{k} L_{GEV}(M^j; \sigma_j, \xi_j)$ `// online GEV` $k$`-means`

    $L_s = -\sum_{j=1}^{k} L_{GPD}(y^j; \sigma_j, \xi_j)$ `// online GPD` $k$`-means`

    Compute the gradient $\bigtriangledown L_s$ and then update the parameters of the GEV or GPD;

**end**

return centroid $\boldsymbol{C}$ ;

---

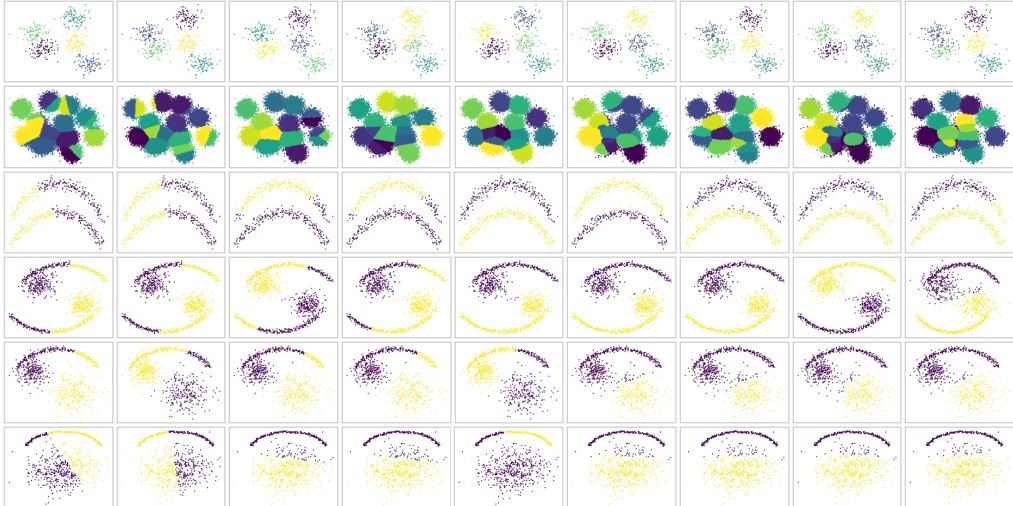

Figure 2: Visualization of six synthetic datasets shows the result of our Extreme Value $k$-means compared to $k$-means, $k$-means++, $k$-medoid, bidecting $k$-means and spectral clustering. The results from top to down are the clustering results on the datasets D1, D2, D3, D4, D5 and D6, respectively. The nine algorithms from the first column to the ninth column are respectively $k$-means, $k$-means++, $k$-medoid, bidecting $k$-means, spectral clustering, GEV $k$-means (RM), GEV $k$-means (++), GPD $k$-means (RM), GPD $k$-means (++).

in the range of 0 to 1, with 1 indicating the best clustering and 0 indicating the worst clustering. The value of ARI is in range of -1 to 1, -1 indicates the worst clustering, and 1 indicates the best clustering.

## 5.2    SYNTHETIC DATASET DEMONSTRATION

We demonstrate our algorithm compared to other algorithms on six two-dimensional synthetic datasets. As illustrated in Fig. 2, there are the clustering results of the datasets D1, D2, D3, D4, D5 and D6 from top to down. D1 consists of 5 isotropic Gaussian clusters, each of which has 100 samples. D2 consists of 15 isotropic Gaussian clusters, each of which has 4000 samples. D3 consists of two 'C'-shaped clusters in the same direction, each of which has 250 samples. D4 consists of two clusters, each of which has 500 samples including a Gaussian blob and a 'C'-shaped region. D5 consists of a Gaussian cluster having 500 samples and a 'C'-shaped cluster having 250 samples. The difference between D5 and D4 is that the lower cluster in D5 has no 'C'-shaped region, and Gaussian blobs has a larger variance. In D5, the upper cluster has 500 samples and the lower cluster has 250 samples. The centroids of GEV $k$-means and GPD $k$-means can be initialized randomly or using $k$-means++. Let 'RM' and '++' denote randomly and using $k$-means++ initialize centroids, respectively. Therefore, there are nine algorithms in this experiment, $k$-means, $k$-means++, $k$-medoid (Kaufman & Rousseeuw, 2009), bisecting $k$-means (Steinbach et al., 2000), spectral clustering (Ng et al., 2002), GEV $k$-means (RM), GEV $k$-means (++), GPD $k$-means (RM), GPD $k$-means (++), respectively. From the clustering results of the nine algorithms on six different synthetic datasets in Fig. 2 , it can be seen that our four algorithms can successfully cluster convex and non-convex clusters, but the clustering results of $k$-means and $k$-means++ on non-convex but visibly well-separated clusters are completely unsuccessful. In addition, the clustering results $k$-medoid, bisecting $k$-means, spectral clustering on D3, D4, D5 is worse than our four algorithms.

## 5.3    REAL DATASET EXPERIMENT

We evaluated the proposed EV $k$-means on nine real datasets: iris ($n = 150, d = 4, k = 3$), breast cancer ($n = 683, d = 10, k = 2$), live disorders ($n = 145, d = 5, k = 2$), heart ($n = 270, d = 13, k = 2$), diabetes ($n = 768, d = 8, k = 2$), glass ($n = 214, d = 9, k = 6$), vehicle ($n = 846, d = 18, k = 4$),

Table 1: Results of online Extreme Value $k$-means on streaming data

| Algorithm | ACC | ARI | NMI | ACC | ARI | NMI | ACC | ARI | NMI |
|---|---|---|---|---|---|---|---|---|---|
| | | iris | | | breast cancer | | | liver disorders | |
| $k$-means | 0.8106 | 0.6053 | 0.6539 | 0.9095 | 0.6679 | 0.5507 | 0.5517 | 0.1233 | 0.1006 |
| $k$-means++ | 0.7613 | 0.5721 | 0.6443 | 0.9105 | 0.6714 | 0.5550 | **0.7069** | **0.1447** | **0.1143** |
| $k$-medoids | 0.8040 | 0.6261 | **0.6869** | 0.8395 | 0.5250 | 0.4481 | 0.6214 | 0.0685 | 0.0446 |
| bisecting $k$-means | 0.8040 | 0.6261 | **0.6869** | 0.8395 | 0.5250 | 0.4481 | 0.6214 | 0.0685 | 0.0446 |
| spectral clustering | **0.8453** | **0.6323** | 0.6678 | **0.9367** | **0.7608** | **0.6628** | 0.7034 | 0.1383 | 0.1090 |
| GEV $k$-means (RM) | 0.6719 | 0.4401 | 0.2916 | 0.7512 | 0.2582 | 0.2632 | 0.6605 | 0.0711 | 0.5279 |
| GEV $k$-means (++) | 0.6610 | 0.4302 | 0.1939 | 0.7587 | 0.2773 | 0.2764 | 0.6619 | 0.0751 | 0.3024 |
| GPD $k$-means (RM) | 0.7960 | 0.5937 | 0.6637 | 0.8582 | 0.5278 | 0.4520 | 0.6582 | 0.099 | 0.0779 |
| GPD $k$-means (++) | 0.8036 | 0.6007 | 0.6667 | 0.8604 | 0.5309 | 0.4543 | 0.6623 | 0.1017 | 0.0801 |
| | | heart | | | diabetes | | | glass | |
| $k$-means | **0.8174** | **0.4011** | 0.3139 | 0.6591 | **0.0992** | **0.0608** | 0.4336 | 0.1517 | 0.2905 |
| $k$-means++ | 0.7841 | 0.3481 | 0.2712 | 0.6473 | 0.0675 | 0.0422 | **0.4505** | 0.1641 | **0.3117** |
| $k$-medoids | 0.7741 | 0.3014 | 0.2443 | 0.6194 | 0.0584 | 0.0427 | 0.4126 | 0.1422 | 0.2705 |
| bisecting $k$-means | 0.7741 | 0.3014 | 0.2443 | 0.6194 | 0.0584 | 0.0427 | 0.4126 | 0.1422 | 0.2705 |
| spectral clustering | 0.8130 | 0.3895 | 0.3053 | 0.6471 | 0.0119 | 0.0034 | 0.4131 | 0.1795 | 0.3010 |
| GEV $k$-means (RM) | 0.6866 | 0.1906 | 0.6466 | 0.6586 | 0.0438 | 0.0171 | 0.4121 | 0.1166 | 0.1625 |
| GEV $k$-means (++) | 0.6990 | 0.2174 | **0.7719** | 0.6583 | 0.0475 | 0.0171 | 0.4222 | 0.1246 | 0.2277 |
| GPD $k$-means (RM) | 0.7865 | 0.3412 | 0.2866 | 0.6516 | 0.0874 | 0.0570 | 0.4313 | **0.1798** | 0.3075 |
| GPD $k$-means (++) | 0.7891 | 0.3464 | 0.2920 | **0.6595** | 0.0928 | **0.0608** | 0.4288 | 0.1783 | 0.3072 |
| | | vehicle | | | MNIST | | | CIFAR10 | |
| $k$-means | 0.3452 | 0.0605 | 0.0901 | 0.7359 | 0.7225 | 0.8350 | 0.6502 | 0.6911 | 0.8464 |
| $k$-means++ | 0.3686 | 0.0803 | 0.1179 | 0.8354 | 0.8242 | 0.8175 | 0.9710 | 0.9740 | 0.9909 |
| $k$-medoids | 0.3642 | 0.0783 | 0.1135 | 0.7808 | 0.7422 | 0.8525 | 0.7239 | 0.6947 | 0.8948 |
| bisecting $k$-means | 0.3642 | 0.0783 | 0.1135 | 0.7808 | 0.7422 | 0.8525 | 0.7239 | 0.6947 | 0.8948 |
| spectral clustering | **0.4184** | **0.1188** | **0.1824** | **0.9854** | **0.9679** | **0.9566** | 0.7092 | 0.6442 | 0.8519 |
| GEV $k$-means (RM) | 0.3430 | 0.0582 | 0.1465 | 0.6228 | 0.5429 | 0.6337 | 0.5532 | 0.5028 | 0.9460 |
| GEV $k$-means (++) | 0.3362 | 0.0593 | 0.1375 | 0.6291 | 0.5584 | 0.6195 | 0.6954 | 0.6449 | 0.7409 |
| GPD $k$-means (RM) | 0.3452 | 0.0640 | 0.1078 | 0.8962 | 0.8739 | 0.9054 | 0.7816 | 0.7953 | 0.9248 |
| GPD $k$-means (++) | 0.3474 | 0.0657 | 0.1113 | 0.9296 | 0.9081 | 0.9227 | **0.9883** | **0.9887** | **0.9960** |

MNIST and CIFAR10. The first seven datasets are available from LIBSVM Data website [1]. MNIST is a dataset comprises 60,000 training gray-scale images and 10,000 gray-scale images of handwritten digits 0 to 9. Each of the training images is represented by an 84-dimensional vector obtained by LeNet (LeCun et al., 1998). So the MNIST dataset we use has 60,000 samples with 84 features belonging to 10 classes, i.e., $n = 60,000, d = 84, k = 10$. CIFAR10 is a dataset containing 50,000 taining and 10,000 test color images with $32 \times 32$ pixels, grouped into 10 different classes of equal size, representing 10 different objects. Each of the training images is represented by a 512-dimensional vector extracted by a ResNet-18 (He et al., 2016). Therefore, the CIFAR10 we use in the experiment has 50,000 samples with 512 features grouped in 10 classes, i.e., $n = 50,000, d = 84, k = 10$.

We repeat each experiment 10 times with different random seeds and took the mean of the results of 10 times experiments as the final result. In each of the experiments, all algorithms that initialize centroids randomly or by using $k$-means++ start from the same initial centroids. The results of EV $k$-means on real datasets are shown in Tab. 1. As shown in Tab. 1, our proposed EV $k$-means on some datasets are comparable to other algorithms, and outperform other algorithms on MNIST and CIFAR10.

## 5.4 STREAMING DATASET EXPERIMENT

We compare online EV $k$-means with $k$-means, $k$-means++, Mini Batch $k$-means (RM) and Mini Batch $k$-means (++) on MNIST and CIFAR10. As illustrated in Tab. 2, the values of the three metrics of online EV $k$-means are slightly smaller than the values of EV $k$-means. However, the values of the three metrics of Mini Batch $k$-means are much smaller than the values of $k$-means. For example, the values of the three metrics of Mini Batch $k$-means on MNIST are 10%, 17%, 8% smaller than the values of $k$-means. However, the values

---

[1]https://www.csie.ntu.edu.tw/ cjlin/libsvmtools/datasets/

Table 2: Results of online Extreme Value $k$-means on streaming data

| Algorithm | ACC | ARI | NMI | ACC | ARI | NMI |
|---|---|---|---|---|---|---|
| | | MNIST | | | CIFAR10 | |
| $k$-means | 0.8276 | 0.8164 | 0.8867 | 0.7168 | 0.7153 | 0.8966 |
| $k$-means++ | 0.8583 | 0.8361 | 0.8961 | 1.0000 | 1.0000 | 1.0000 |
| Mini Batch $k$-means (RM) | 0.7465 | 0.6759 | 0.8167 | 0.6091 | 0.5137 | 0.8355 |
| Mini Batch $k$-means (++) | 0.8244 | 0.8018 | 0.8798 | 1.0000 | 1.0000 | 1.0000 |
| GEV $k$-means (RM) | 0.9751 | 0.9454 | 0.9344 | 1.0000 | 1.0000 | 0.9460 |
| GEV $k$-means (++) | 0.9758 | 0.9469 | 0.9370 | 1.0000 | 1.0000 | 1.0000 |
| GPD $k$-means (RM) | 0.9792 | 0.9543 | 0.9418 | 0.8940 | 0.8916 | 0.9644 |
| GPD $k$-means (++) | 0.9794 | 0.9547 | 0.9420 | 1.0000 | 1.0000 | 1.0000 |
| online GEV $k$-means (RM) | 0.9375 | 0.8705 | 0.8832 | 0.8505 | 0.8511 | 0.9407 |
| online GEV $k$-means (++) | 0.8478 | 0.8255 | 0.8920 | 1.0000 | 1.0000 | 1.0000 |
| online GPD $k$-means (RM) | 0.9530 | 0.8987 | 0.8921 | 0.8568 | 0.8627 | 0.9400 |
| online GPD $k$-means (++) | 0.9669 | 0.9286 | 0.9156 | 1.0000 | 1.0000 | 1.0000 |

of the three metrics of online GVE $k$-means (RM) on MNIST are 4%, 8%, 5% smaller than the values of GVE $k$-means (RM).

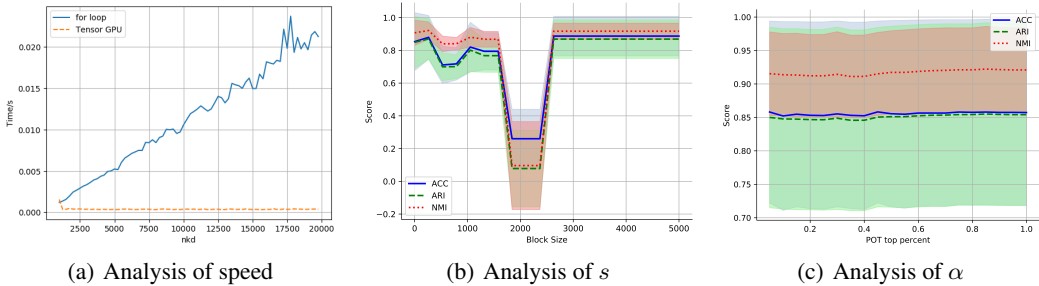

(a) Analysis of speed      (b) Analysis of $s$      (c) Analysis of $\alpha$

Figure 3: On a dataset of 5 Gaussian clusters with 5000 samples, we analyze our acceleration method, block size $s$ and the percentage of excess $\alpha$.

### 5.5 ABLATION STUDY

In Fig. 3(a), 'for loop' means that the Euclidean distance is computed on the CPU using a double-layer nested for loop, and 'Tensor GPU' indicates the use of the acceleration method we proposed. As shown in Fig. 3(a), the computational time using the 'for loop' method increases linearly with the increase of $nkd$. Compared to 'for loop', using the 'Tensor GPU' method can significantly accelerate the computation of Euclidean distance for the computational time is almost unchanged with the increase of $nkd$. 3(b) shows that as the block size increases, ACC, ARI, and NMI both show a sharp drop first, then gradually rise, and finally remain steady. This confirms the Theorem 3.1, the block size should be large enough, then the distribution function $F$ can approximate the GEV distribution. In the application, this paper suggests that the block size $s > \frac{n}{2}$. As shown in 3(c), as $\alpha$ increases, ACC, ARI, and NMI show a steady trend. In order to meet the conditions of Theorem 3.2, $\alpha$ should be relatively small to get a large enough $u$. In the experiment of this paper, we set $\alpha$ to 0.1.

## 6 CONCLUSIONS

This paper introduces Extreme Value Theory into $k$-means to measure similarity by transforming the Euclidean space into extreme value space. Based on the strong theoretical support provided by EVT, this paper proposes Extreme Value $k$-means, a novel algorithm for the task of $k$-means clustering. In view of the bottleneck of $k$-means, this paper proposes a practical method to accelerate the computation of Euclidean distance. As for streaming data clustering, this paper presents online Extreme Value $k$-means that can perform clustering streaming data. We evaluate the performance of our algorithm on synthetic datasets and real datasets. The results show that our algorithm outperform $k$-means and $k$-means++ on all datasets.

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
