# OpenReview forum: "Extreme Value k-means Clustering"
_ICLR.cc/2020/Conference — Reject_

### Official Review · AnonReviewer1 · 2019-10-22
**Official Blind Review #1**

**Rating:** 3

**Review:**

In this paper, the authors propose an improved k-means clustering algorithm by using a new similarity measurement method based on extreme value theory. Extreme value theory can better capture the data distribution and thus better handle non-convex situations. Experiments on extensive datasets are conducted to show the effectiveness of the proposed method.

The paper is well written and easy to understand. However there are some concerns:
1. There are many clustering methods based on geodesic distance of data points (manifold learning), which are supposed to be better at capturing the non-convex data distribution. Comparisons are suggested.
2. The datasets used in the experiments are relatively small (n<=50000, and k<=10), what about more clusters?
3. The results of different methods may be unstable to initialization, more tests and average results are suggested.



**Experience Assessment:**

I have read many papers in this area.

**Review Assessment: Checking Correctness Of Derivations And Theory:**

I assessed the sensibility of the derivations and theory.

**Review Assessment: Checking Correctness Of Experiments:**

I assessed the sensibility of the experiments.

**Review Assessment: Thoroughness In Paper Reading:**

I read the paper at least twice and used my best judgement in assessing the paper.

---

### Official Review · AnonReviewer2 · 2019-10-22
**Official Blind Review #2**

**Rating:** 1

**Review:**

The paper claims to improve the “clustering ability” of k-means by measuring the similarity between samples and centroids by something known as Extreme Value Theory (EVT). The paper in its current form is extremely difficult to follow. I highlight some of the difficulty:

1. The notion of “clustering ability” of k-means is not well defined in the paper. So, in general there does not seem to be a clear theoretical objective that is being optimised by the suggested algorithm.

2. The algorithm itself is not clear. For example, the block maximum sequence M^j = {M_1, …, M_m} is not clearly understood. What is threshold u_j?

Given the above, I am unable to evaluate the contribution of the paper. It will help the paper if the authors clearly formalise what is it that they are trying to achieve with their algorithm. After this, there needs to be a much more clear description of the algorithm. I think the above will also help decide whether section 3.2 on Extreme Value Theory is really required for the discussion.


**Experience Assessment:**

I have published in this field for several years.

**Review Assessment: Checking Correctness Of Derivations And Theory:**

I assessed the sensibility of the derivations and theory.

**Review Assessment: Checking Correctness Of Experiments:**

I assessed the sensibility of the experiments.

**Review Assessment: Thoroughness In Paper Reading:**

I read the paper at least twice and used my best judgement in assessing the paper.

---

### Official Review · AnonReviewer3 · 2019-10-29
**Official Blind Review #3**

**Rating:** 6

**Review:**

This paper utilizes EVT to improve k-means, which aims to address the problem of clustering data of nonconvex shape. GEV k-means and GPD k-means are proposed as two kind of  Extreme Value k-means. A method for accelerating Euclidean distance computation has also been proposed to solve the bottleneck of k-means.
The proposed idea is novel and the paper is well written. Experimental reaultts are also good to me.
There are two concerns which I need the authors to address:
1. Since the authors claim that they propose to speed up the computation of the Euclidean distances in k-means. However, there is no time cost comparison in the experiments.
2. Some other variants of k-means should be added for experimental comparison.

**Experience Assessment:**

I have published in this field for several years.

**Review Assessment: Checking Correctness Of Derivations And Theory:**

I assessed the sensibility of the derivations and theory.

**Review Assessment: Checking Correctness Of Experiments:**

I carefully checked the experiments.

**Review Assessment: Thoroughness In Paper Reading:**

I read the paper thoroughly.

---

### Official Review · AnonReviewer4 · 2019-11-03
**Official Blind Review #4**

**Rating:** 3

**Review:**

The paper considers extending the k-means algorithm to allow for finding clusters with non-convex shapes. Particularly, it uses an existing theoretical framework (Extreme Value Theory) to maps a Euclidean space into what it calls the extreme value space, and proposes two Extreme Value k-means algorithms: GEV k-means and GPD k-means. It then provides some empirical results demonstrating their approach.

I have some concerns with the paper's claimed novelties, its empirical evaluation, and its overall presentation, and thus am initially recommending a weak reject. To raise my score, the following concerns should be clarified or addressed:

1) An initial concern is with the claimed novelty of the work. A paper by Li et al. (2012) also uses Extreme Value Theory (EVT) to improve k-means using the Generalized Extreme Value (GEV) distribution. Their algorithm is also called GEV k-means, and is based on an observation that the squared distance from a point to its closest center follows the GEV distribution for large numbers of clusters. From this, it doesn't appear to be the first time that EVT has been used to improve k-means, and it would be good for the authors to contrast their methods in the context of existing work in this direction.

2) Is it generally applicable to measure similarity based on the probability of being an extreme value, compared with classic metrics like Euclidean distance? In other words, would it always be better to do this or are there clear counterexamples where you would not want to measure similarity based on this?

3) In the empirical evaluation, it is said that 10 independent runs were performed, and the maximum result of the 10 runs was reported. I believe it would be more informative to report the mean, or expected performance of the algorithm, as well as some statistic about the mean to ensure any differences are significant. It is not clear whether this maximum can be expected or reproduced, and can negatively be interpreted as the algorithm having considerably higher variability- in other words, it could be the case that the minimum of the 10 runs for the EV methods was also lower than the minimums of classic k-means. Were any statistical tests done to ensure that the larger maximum over the runs was not by chance?

4) The paper has many frequent, but minor, grammatical and spelling errors. As such, it is possible to get the overall message (and didn't strongly impact my score), but it does detract from the paper's overall presentation and quality.

-----

Post-rebuttal:

Thank you for your clarifications regarding 1) and 2), and for now reporting the mean. However, there still does not seem to be any statistical significance testing. Further, after reporting the mean and comparing with more methods, the method doesn't seem to perform as well as previously reported. This makes the concern in 2) more prevalent- if the method is not generally applicable, and are likely to help on a problem-specific basis, it would be more informative to characterize *when* one might expect the method to perform better, and support this with empirical results. Based on this, I am maintaining my score, but think the work is interesting and encourage the authors to improve on the paper!

**Experience Assessment:**

I do not know much about this area.

**Review Assessment: Checking Correctness Of Derivations And Theory:**

I assessed the sensibility of the derivations and theory.

**Review Assessment: Checking Correctness Of Experiments:**

I assessed the sensibility of the experiments.

**Review Assessment: Thoroughness In Paper Reading:**

I read the paper at least twice and used my best judgement in assessing the paper.

---

### Decision · Program_Chairs · 2019-12-19

**Decision:**

Reject

**Comment:**

This paper explores extending k-means to allow to clusters with non-convex shapes.

This paper introduces a new algorithm, relying on empirical comparisons to illustrate its contribution. The main issue with the paper is that the empirical claims do not support that the new method is indeed better. The paper claims the new method outperforms the competitors in most cases. However, the original submission reported median performance and when the authors provided mean performance and additional baseline methods (at the reviewers' request) there appear to be little evidence to support the claim. In addition there are no measures of significance provided. The authors provided no commentary to help the reviewers understand the new results. There might be some important speed gains at the cost of final performance, but on the evidence provided we are not able to evaluate the cost in final performance.

The text changes size after section 5.3 and is 9% smaller. Watch out for this formatting issue in future submissions